# Evaluation of serum VIP and aCGRP during pulmonary exacerbation in cystic fibrosis: A longitudinal pilot study of patients undergoing antibiotic therapy

Maha S. Al-Keilani[1]*, Samah Awad[2,3], Hanan M. Hammouri[4], Tala Al Shalakhti[2], Basima A. Almomani[1], Muna M. Dahabreh[5], Mohammad-Jaafar Ajlony[6]

1 Department of Clinical Pharmacy, College of Pharmacy, Jordan University of Science and Technology, Irbid, Jordan, 2 Department of Pediatrics and Neonatology, College of Medicine, Jordan University of Science and Technology, Irbid, Jordan, 3 Department of Pediatrics, College of Medicine, University of Arkansas for Medical Sciences, Little Rock, Arkansas, United States of America, 4 Department of Mathematics and Statistics, College of Science and Arts, Jordan University of Science and Technology, Irbid, Jordan, 5 Department of Respiratory Medicine, Royal London Hospital Barts NHS Trust, London, United Kingdom, 6 Department of Pediatrics, Princess Rahma Teaching Hospital, Irbid, Jordan

* mskeilani@just.edu.jo

## Abstract

### Background

Objective monitoring of improvement during treatment of pulmonary exacerbation can be difficulty in children when pulmonary function testing cannot be obtained. Thus, the identification of predictive biomarkers to determine the efficacy of drug treatments is of high priority. The major aim of the current study was to investigate the serum levels of vasoactive intestinal peptide (VIP) and alpha calcitonin gene related peptide (aCGRP) of cystic fibrosis pediatric patients during pulmonary exacerbation and post-antibiotic therapy, and possible associations of their levels with different clinicopathological parameters.

### Methods

21 patients with cystic fibrosis were recruited at onset of pulmonary exacerbation. Serum was collected at time of admission, three days post-antibiotic therapy, and two weeks post-antibiotic therapy (end of antibiotic therapy). Serum VIP and aCGRP levels were measured using ELISA.

### Results

Overall least square means of serum aCGRP level but not VIP changed from time of exacerbation to completion of antibiotic therapy (p = 0.005). Serum VIP was significantly associated with the presence of diabetes mellitus (p = 0.026) and other comorbidities (p = 0.013), and with type of antibiotic therapy (p = 0.019). Serum aCGRP level was significantly associated with type of antibiotic therapy (p = 0.012) and positive *Staphylococcus aureus* microbiology test (p = 0.046).

**Data Availability Statement:** All relevant data are within the manuscript and its Supporting information files.

**Funding:** MSA received the fund from the Deanship of Scientific Research at Jordan University of Science and Technology, Irbid, Jordan. https://www.just.edu.jo/Deanships/DeanshipofResearch/Pages/Default.aspx Research Grant No: 20180020. The funders had no role in study design, data collection and analysis, decision to publish, or preparation of the manuscript.

**Competing interests:** The authors have declared that no competing interests exist.

## Conclusion

This study could only show significant changes in serum aCGRP levels following treatment of pulmonary exacerbations. Future studies with larger sample size are required to investigate the clinical importance of VIP and aCGRP in cystic fibrosis patients.

## Introduction

Cystic fibrosis (CF) is the most common fatal genetic disease among Caucasians. Impairments in water and salt secretion, decreased fluid volume, increased mucus thickness, and viscosity [1, 2]. Additionally, submucosal glands undergo subsequent changes that expand the previous effects [2]. Consequently, a multisystem inflammatory condition results, with multifaceted pulmonary inflammation being the most common. The subsequent progressive lung destruction and decline in lung function with resultant respiratory failure are the major contributors of increased morbidity and mortality [3]. CF is heterogeneous at clinical level in terms of disease manifestations, progression and occurrence of pulmonary exacerbations [4]. This highlights the importance of identifying biomarkers that aid in the early diagnosis of pulmonary exacerbations, better prediction of disease progression, and monitoring response to therapy. Moreover, they could serve as new therapeutic targets for CF.

The only currently available objective parameter for classifying severity of CF and monitoring response of patients to therapy is the clinically based pulmonary function testing, FEV1 (forced exhaled volume at first second) [5]. However, such testing is not feasible in young children and in patients with severe lung disease and is not accurate in mild conditions where the test is usually normal [6]. Thus, the identification of biomarkers that are easily measurable in patients of any age and disease severity and that allow the early follow-up of patients is vital.

Neuropeptides, immunomodulatory proteins that are distributed throughout the peripheral and central nervous systems, are among the potential candidates. Neuropeptides are involved in the neuro-immune communication and play major roles in inflammation through regulating cytokine production. Vasoactive intestinal peptide (VIP) and alpha calcitonin gene-related peptide (αCGRP) are among the neuropeptides that have potential roles in CF.

VIP is the most abundant neuropeptide in the lung, and experimental work revealed many functions including bronchodilation, increased mucus secretion from airway glands and goblet cells, vasodilation, besides the anti-inflammatory and immunomodulatory effects [7]. In a study by Mandal, et al on COPD patients, acute exacerbations were associated with significantly higher serum VIP levels as compared to stable condition [8]. Moreover, serum VIP levels were significantly higher in children with asthma as compared to controls [9].

aCGRP is a 37 amino acid vasodilatory neuropeptide that is distributed to the central and peripheral nervous systems and is localized to the airway submucosal glands and neuroendocrine cells [10]. It exerts many functions on airways including bronchial protection [11], regulation of macrophages function and anti-inflammation [12], regulation of airway responsiveness [13], proliferation of tracheal and alveolar epithelial cells [14], and induction of mucus secretion from submucosal glands and goblet cells [2]. When compared to controls, serum aCGRP levels were significantly higher in pediatric patients with bronchial asthma [15], and significantly lower in COVID19 patients [16].

In this study we sought to identify VIP and aCGRP as blood biomarkers that can be easily assayed and are still applicable and reliable for predicting response to antibiotic therapy in CF patients after pulmonary exacerbations. As potential blood biomarkers, VIP and aCGRP may

be considered of great advantage since they would be easily standardized, may link between pulmonary and non-pulmonary complications of CF, and they may acutely change in response to treatment, thus providing rapid insight into therapeutic outcomes.

## Methods

### Study population and design

Patients with CF were recruited prospectively when diagnosed clinically by a pulmonologist to have pulmonary exacerbation and were admitted to Princess Rahma Teaching Hospital (PRTH), King Abdullah University Hospital (KAUH) and Royal Medical Services (RMS), Jordan. The study period lasted about 1 year (July, 2018 until August, 2019). Pulmonary exacerbation was defined if there was worsening in respiratory symptoms and signs beyond the usual day-to-day variation. All patients had at least 4 of the 12 Fuchs criteria which include: an increased cough and expectoration, purulent sputum, hemoptysis, loss of appetite and weight, dyspnea, fever, decrease of >10% in FEV1 and/or oxygen saturation with respect to baseline values, changes in lung sounds, and radiographic evidence of new infiltrates [17]. Transplant patients or patients receiving chronic immunosuppressive therapy were excluded from the study. Participating subjects received antibiotic treatment in conjunction with airway clearance therapies such as physiotherapy, mucolytics and bronchodilators. The duration of antibiotic therapy for pulmonary exacerbation was 2 weeks for all patients, which is the duration of antibiotic therapy where patients returned to a symptomatic baseline and had clinically improved based on clinician's assessment. On the day of admission, we recorded the following clinical variables: gender, age, body mass index (BMI), CF-related diabetes mellitus (defined as the need for insulin to control blood sugar level), CF-related liver disease, sputum microbiology, and type of antibiotic prescribed during admission. BMI z-scores and percentiles were calculated using Centers for Disease Control and Prevention (CDC) calculator [18].

### Serum sample collection and ELISA measurements

Five ml of peripheral blood were collected in tubes containing protease inhibitors (aprotinin, EDTA) and then centrifuged at 1,600 x g for 15 minutes at 4˚C to isolate and collect serum. Serum samples were kept at -80˚C till analysis. A blood sample was collected in three occasions, at hospital admission (VIP1, aCGRP1) before receiving antibiotic therapy, three days post-antibiotic (VIP2, aCGRP2), and on the day of discharge (2 weeks post-antibiotic; VIP3, aCGRP3). The Serum levels of VIP (IPR015523, CLOUD-CLONE CORP, USA, cat. # CEA380Hu) and aCGRP (IPR015476, CLOUD-CLONE CORP, USA, cat. # CEA472Hu) were measured using human enzyme-linked immunosorbent assay (ELISA) technique. The absorbance was measured at the appropriate wavelength using ELx800 plate reader (Bio-teak instruments, Winooski, USA). Serum levels of the proteins are expressed in pg/ml.

### Ethical approval

The research was performed in accordance with the Declaration of Helsinki, and the protocol was approved by the Research and Ethics Committee at Jordan University of Science and Technology (registration number 34/111/2017, date of approval January 7, 2018), where all methods were performed in accordance with the relevant guidelines and regulations.

### Consent to participate

All patients signed a written informed consent at the start of the study. For patients under 18-years-old, the informed consent was signed by one of their parents.

## Statistical analysis

Data were presented as number (percent) for categorical variables and mean (SEM) for continuous variables. Q-Q plots were used to verify the normality of VIP and aCGRP variables. There was a need for log transformations to normalize VIP variables, whereas aCGRP variables could be assumed to be normally distributed without requiring any transformations. By analyzing the changes overtime of LOG VIP and aCGRP2 serum levels using repeated measures models, we investigated the factors responsible for these changes. Forest plots for means and 95% confidence intervals (95% CI) were used to display treatment effects across the three time points and to check the univariate effect of different covariates. Repeated measures multiple ANOVA (MANOVA) was used to determine whether there are any differences in multiple dependent variables over time using the backward selection model. Statistical analysis was performed utilizing SPSS statistical software system (IBM SPSS Statistics 23, USA) and JMP Pro®, Version 16 (SAS Institute Inc., USA). $P \leq 0.05$ was considered statistically significant.

## Results

A total of 21 patients were recruited in the study. Clinical characteristics of participating subjects are summarized in Table 1. In brief, the mean age of patients was 8.71 years (SEM 1.02), and 14 patients (66.7%) were males. The mean BMI z-score of patients was -2.09 (SEM 0.61). Nine patients had comorbidities including: asthma (n = 1), convulsions (n = 1), hypothyroidism (n = 1), celiac disease (n = 1), diabetes mellitus and osteoporosis (n = 1), diabetes mellitus only (n = 1), and liver disease (n = 3). Nine patients (42.9%) were positive for *P. aeruginosa*, and 15 patients (71.4%) were positive for *S. aureus*, one of them was MRSA. Nine patients (42.8%) were prescribed ciprofloxacin, 5 patients (23.8%) were prescribed azithromycin, and the remaining received amoxicillin/clavulanic acid, cefixime, gentamicin, or a combination of azithromycin with amoxicillin/clavulanic acid. Serum levels of VIP and aCGRP were measured at hospital admission (VIP1, aCGRP1) before receiving antibiotic therapy, three days post-antibiotic (VIP2, aCGRP2), and on the day of discharge (2 weeks post-antibiotic; VIP3, aCGRP3).

As shown in Fig 1, which reveals the LS means, it was found that there were significant between-subjects effects over the LOG VIP. In terms of overall LS means, there was no significant difference over time (p>0.05; Fig 1A). Statistically significant differences in LOG VIP LS means were found by the presence of diabetes mellitus (p = 0.026; Fig 1B), or other comorbidities (p = 0.013; Fig 1C), and by type of medication (p = 0.019; Fig 1D). Patients with diabetes mellitus or other comorbidities had higher levels than those without. The levels were the highest in patients who received ciprofloxacin and the lowest in patients who received cefixime.

In the aCGRP model, the overall LS means differed significantly (p-value = 0.005; Fig 2A). The LS mean started at 364.5 and increased to 350.2, then decreased slightly to 349.8. Statistically significant differences in aCGRP LS means (between-subject effect) were found by type of medication (p = 0.012; Fig 2B) and by positive S. aureus infection (within-subject effect) (p = 0.046; Fig 2C). Patients who received ciprofloxacin had the highest levels. Patients who had positive S. aureus infection had lower levels than those without the infection. LS mean increased over time for groups with positive S-aureus, followed by a decrease. In the group that did not have infection, the LS mean decreased then increased over time.

To check the univariate different effects over the differences LOG VIP2- LOG VIP1 (Fig 3A), LOG VIP3- LOG VIP1 (Fig 3B), aCGRP2- aCGRP1 (Fig 4A) and aCGRP3- aCGRP1 (Fig 4B), forest plots for means and 95% CI were used. Based on the plots, all 95% CI contained zeros, which indicates that there are no differences between the two readings for each level.

**Table 1. Demographics and clinical characteristics of study subjects.**

| Variable* | |
|---|---|
| **Age, years** | 8.71 ± 1.02 |
| **Gender** | |
| Male | 14 (66.7) |
| Female | 7 (33.3) |
| **BMI** | 15.77 ± 0.85 |
| **BMI z-score** | -2.09 ± 0.61 |
| **BMI percentile** | 24.43 ± 6.72 |
| **BMI category** | |
| Underweight | 9 (42.9) |
| Normal | 11 (52.4) |
| Overweight | 0 (0) |
| Obese | 1 (4.8) |
| **Comorbidities** | |
| No | 12 (57.1) |
| Yes | 9 (42.9) |
| **Diabetes mellitus** | |
| No | 19 (90.5) |
| Yes | 2 (9.5) |
| **Liver disease** | |
| No | 18 (85.7) |
| Yes | 3 (14.3) |
| ***P. aeruginosa* +ve** | 9 (42.9) |
| ***S. aureus* +ve** | 15 (71.4) |
| **MRSA +ve** | 1 (4.8) |
| **Antibiotic prescribed during admission** | |
| Ciprofloxacin | 10 (47.6) |
| Azithromycin ± (Amoxicillin + clavulanic acid) | 8 (38.1) |
| Cefixime | 2 (9.5) |
| Gentamicin | 1 (4.8) |
| **VIP** | |
| At hospital admission (VIP1) | 450.67 ± 171.21 |
| 3 days post-antibiotic therapy (VIP2) | 325.45 ± 87.69 |
| 2 weeks post-antibiotic therapy (VIP3) | 331.30 ± 81.70 |
| **aCGRP** | |
| At hospital admission (aCGRP1) | 346.54 ± 47.19 |
| 3 days post-antibiotic therapy (aCGRP2) | 350.24 ± 43.31 |
| 2 weeks post-antibiotic therapy (aCGRP3) | 349.79 ± 68.26 |

* Data are presented as mean ± SEM or as number (%)

*P. aeruginosa*: *Pseudomonas aeruginosa*

*S. aureus*: *Staphylococcus aureus*

MRSA: Methicillin-Resistant *Staphylococcus Aureus*

VIP: Vasoactive Intestinal Peptide

aCGRP: alpha Calcitonin Gene-Related Peptide

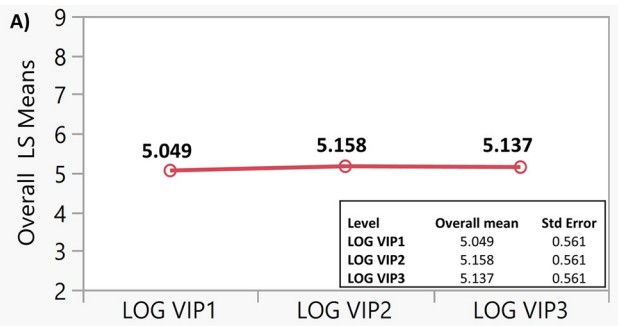
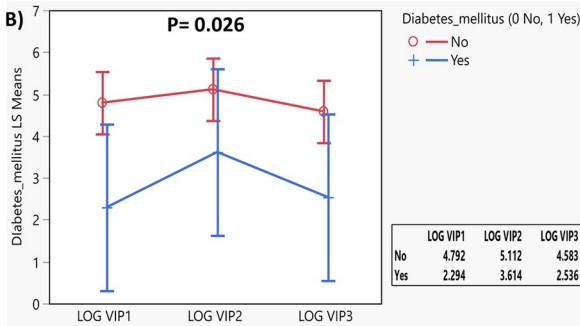
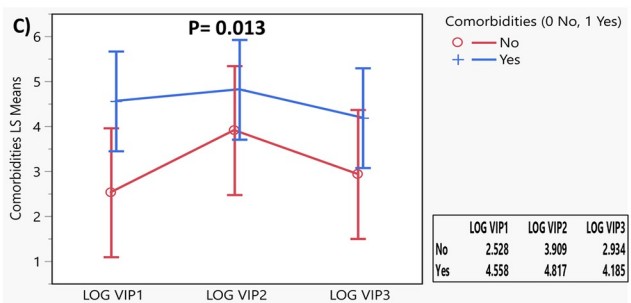
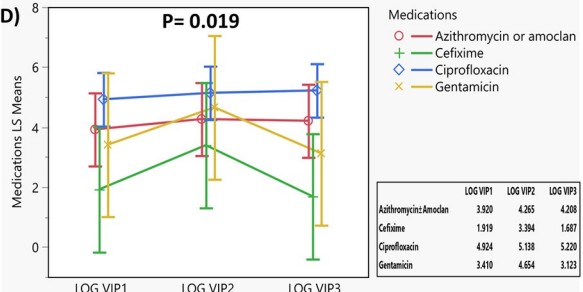

**Fig 1. Least Square (LS) means changes in log serum level of vasoactive intestinal peptide (VIP) in patients with cystic fibrosis with pulmonary exacerbation during baseline (at exacerbation; LOG VIP1), 3-days post-antibiotic (LOG VIP2), and 2 weeks post-antibiotic (LOG VIP3).** A). Overall LS means in LOG VIP. B). LS means in log VIP level by diabetes mellitus. C). LS means in log VIP level by comorbidities. D). LS means in log VIP level by type of medication.

## Discussion

This is the first study to investigate the serum levels of VIP and aCGRP in CF patients and detect the changes in levels in response to antibiotic therapy during the treatment of CF pulmonary exacerbations. In the present study we could demonstrate that serum level of aCGRP but not VIP predicts response to antibiotic therapy in CF patients with pulmonary exacerbations.

Pulmonary exacerbation is one of the most clinically important events in CF patients as it may result in several complications including permanent loss of lung function and increased morbidity and mortality [19]. Diagnosis and treatment of CF acute pulmonary exacerbations vary among clinicians and are highly dependent on the clinical assessment of patients. Thus, a biomarker that potentially diagnoses an acute pulmonary exacerbation and predicts response to therapy would therefore be beneficial for CF management.

Several serum markers have been reported to be changed at CF pulmonary exacerbation, such as caloptrectin [20], Short Palate Lung Nasal epithelium Clone 1 (SPLUNC1) [21], inflammatory markers like IL-8, C-reactive protein and neutrophil elastase antiprotease complexes (NEAPC) [22], and procalcitonin [23]. However, there have been no studies investigating the relationship between serum VIP and aCGRP levels and CF pulmonary exacerbation. Therefore, to our knowledge, this is the first study to analyze the serum VIP and aCGRP levels in CF patients at acute pulmonary exacerbation and post-antibiotic therapy.

Previous reports indicate that VIP and aCGRP may be involved in the pathophysiology of pulmonary exacerbations in CF patients [2, 24–27]. In the current study we could report changes in serum levels of aCGRP but not VIP in CF patients with pulmonary exacerbations

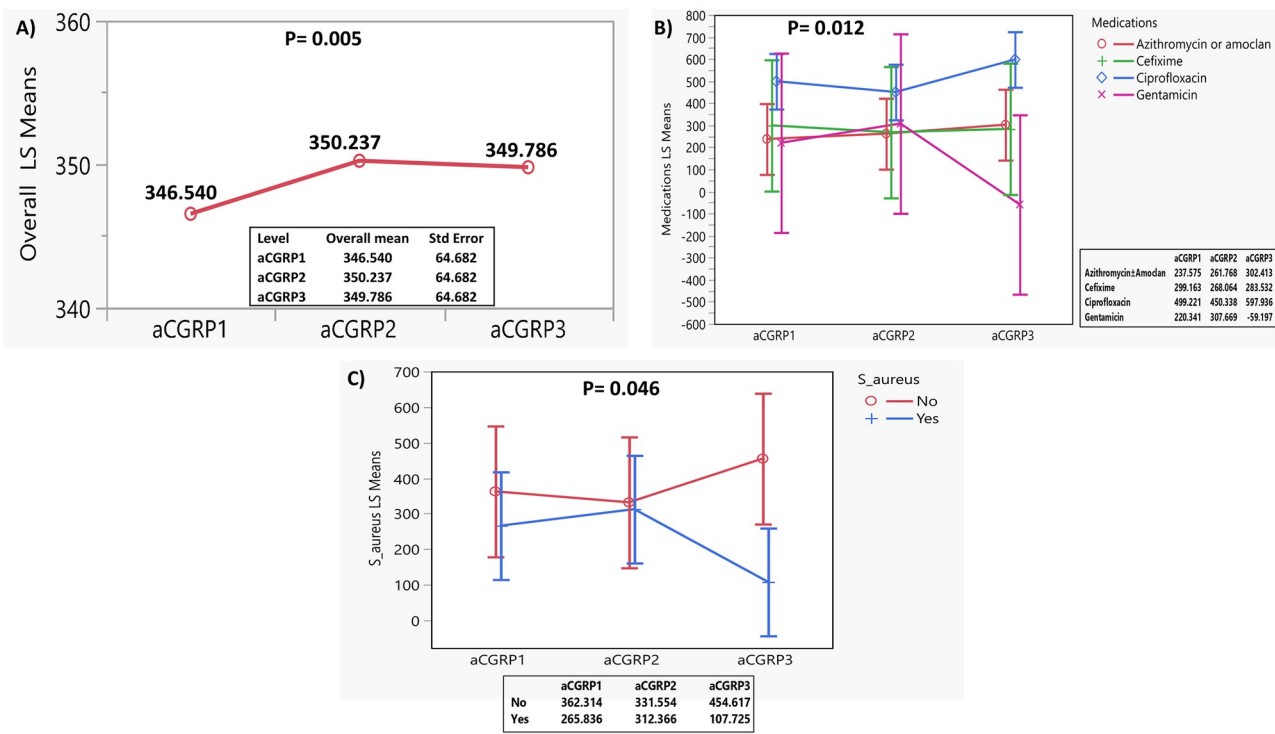

**Fig 2. Least Square (LS) means changes in serum level of alpha calcitonin gene related peptide (aCGRP) in patients with cystic fibrosis with pulmonary exacerbation during baseline (at exacerbation; aCGRP1), 3-days post-antibiotic (aCGRP2), and 2 weeks post-antibiotic (aCGRP3).** A). Overall LS means in aCGRP level. B). LS means in aCGRP level by type of medication. C). LS means in aCGRP level by infection with Staphylococcus aureus (S. aureus).

in response to treatment. Significant between-subjects effects over the LOG VIP indicates that the difference between the levels means is not due to the time.

Vasoactive intestinal peptide (VIP) is a 28-amino acid peptide that exerts its pleiotropic functions via binding to two G-protein-coupled receptors, VPAC1 and VPAC2 that are widely expressed in the peripheral tissues [28, 29]. VIP plays a key role in inflammation, and airway and vascular smooth muscles relaxation [10], and fluid secretion from submucosal glands was stimulated by VIP in normal mice but not in CFTR-knockout mice, suggesting CFTR-dependent mechanisms [30]. VIP modulates inflammatory response via enhancing the production of anti-inflammatory molecules and reducing the pro-inflammatory cytokines and chemokines. A significant association between blood level of VIP and inflammatory diseases was previously shown in several studies [31–33]. Patients with severe coronavirus disease 2019 (COVID-19) had significantly higher plasma levels of VIP as compared to patients with mild COVID-19 symptoms and noninfected healthy individuals, and these high levels were associated with better survival [33]. Consequently, and due to the widespread distribution of VIP in the body including the lungs, VIP is a potential biomarker in CF and may play protective roles during pulmonary exacerbations of CF.

In the current study, we could not show a significant change in serum VIP levels from exacerbation to two-weeks after treatment with antibiotics. However, we found that patients with diabetes mellitus had significantly lower VIP levels than patients without the disease. The role of VIP in diabetes mellitus at molecular level has been reviewed by many researchers [34, 35]. Due to its anti-inflammatory and insulinotropic effects; VIP has been suggested as a potential therapeutic target for treatment of types 1 and 2 diabetes mellitus [36–39].

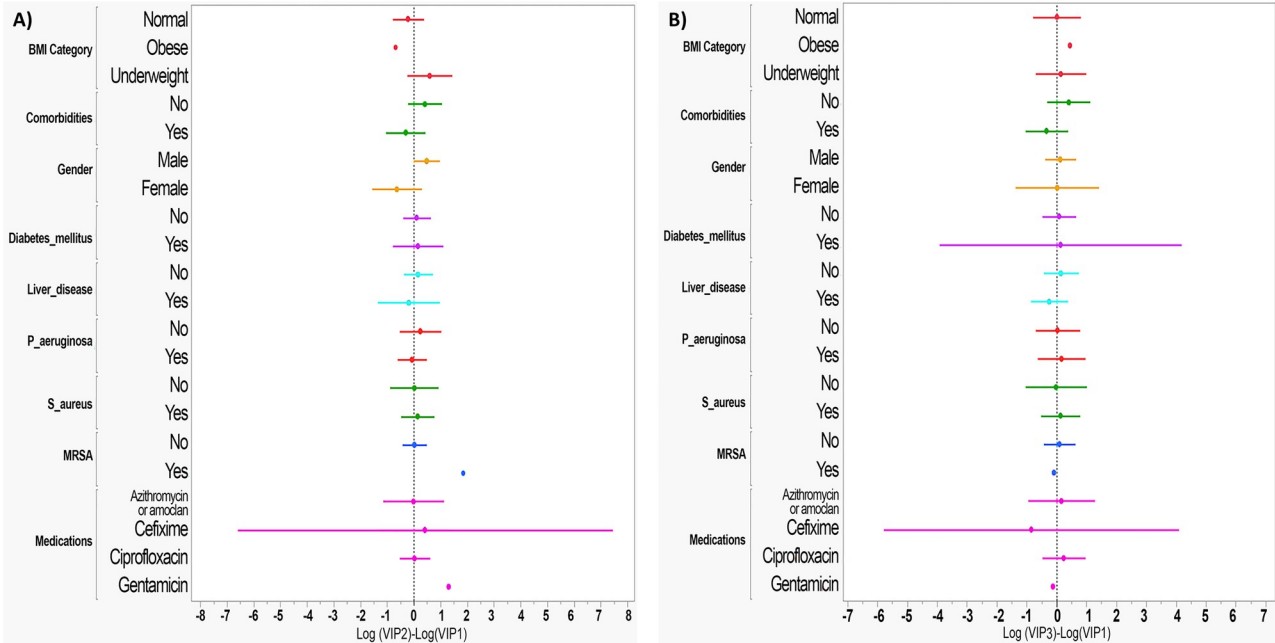

**Fig 3. Forest plot of univariate analysis for factors associated with serum VIP level.** A). Univariate analysis for difference between serum VIP level at baseline (Log VIP1) and 3-days post-antibiotic therapy (Log VIP2). B). Univariate analysis for difference between serum VIP level at baseline (Log VIP1) and 2 weeks post-antibiotic therapy (Log VIP3).

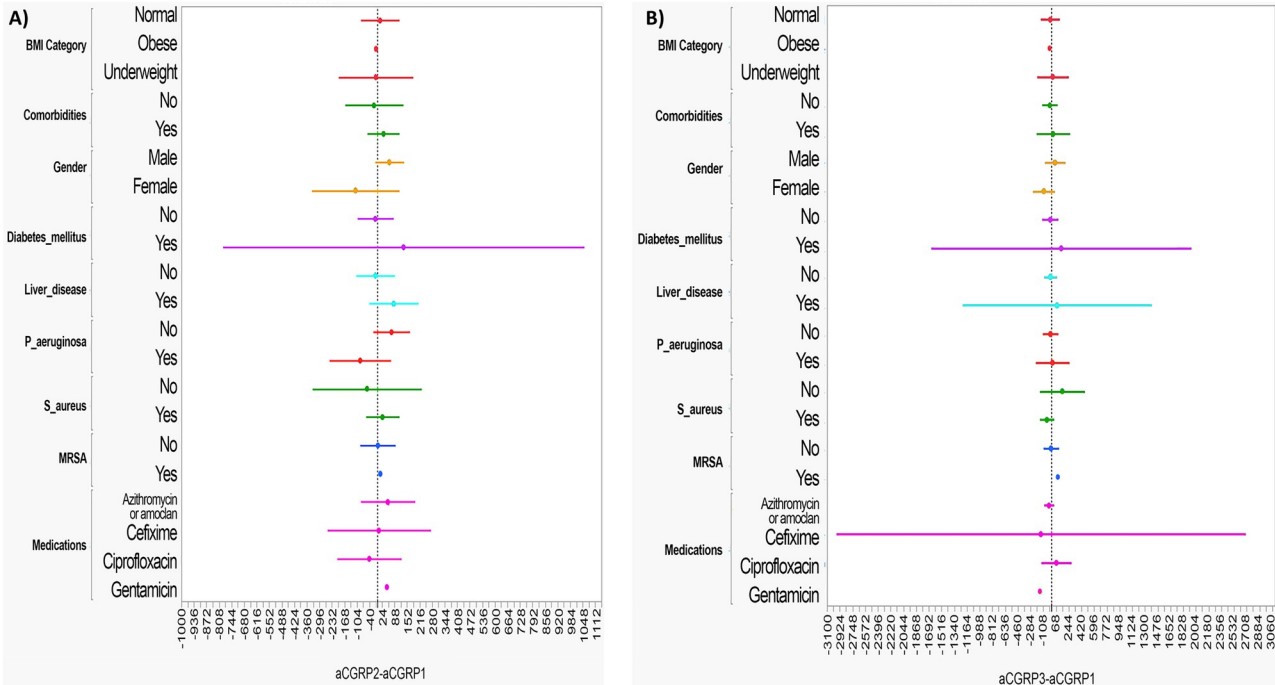

**Fig 4. Forest plot of univariate analysis for factors associated with serum aCGRP level. A).** Univariate analysis for difference between serum aCGRP level at baseline (aCGRP1) and 3-days post-antibiotic therapy (aCGRP2). **B).** Univariate analysis for difference between serum aCGRP level at baseline (aCGRP1) and 2 weeks post-antibiotic therapy (aCGRP3).

Furthermore, in experimental models of diabetes mellitus, diabetic rats had significantly lower gastrointestinal VIP protein levels but higher plasma levels than normal rats [40], which may indicate that VIP leaks from the gastrointestinal tract to the blood. In the current study, due to disproportionate sample size between the diabetic and nondiabetic patients and the absence of a healthy control group, a firm conclusion regarding the differential role of VIP in CF patients with diabetes cannot be withdrawn.

A possible role of aCGRP in CF was also suggested, as it activated CFTR-dependent secretions from submucosal glands and was overexpressed in CF human, pig, ferret, and mice [2, 41]. Additionally, it was suggested to play a key role on CF exacerbations through induction of secretion of muc5AC, a gel-forming glycoprotein of gastric and respiratory tract epithelia that protects the mucosa from infection and chemical damage by binding to inhaled micro-organisms and particles that are subsequently removed by the mucociliary system [10]. In our study we found that the overall LS means of aCGRP level significantly increased over time in response to antibiotic therapy. This significant increase may participate in better response to antibiotic therapy through induction of immunomodulatory and anti-inflammatory responses. Significant airway inflammation developed post-infection with respiratory syncytial virus (RSV), which was associated with significant reduction in the expression level of aCGRP in the airway tissues, and treatment with aCGRP resulted in reduced airway hyper-responsiveness thus suggesting the role of this neuropeptide in airway inflammation [42]. Our results are in parallel with previous studies that demonstrated significant changes in systemic inflammatory markers' levels following antibiotic therapy during exacerbations [22, 43, 44]. Altogether, these findings support our hypothesis that aCGRP is a potential biomarker for pulmonary exacerbation and response to therapy and is a promising therapeutic target in CF patients.

Treatment with some antibiotics was associated with changes in serum levels of VIP and aCGRP post-antibiotic therapy which indicates that these antibiotics exert immunomodulatory effects on VIP- and aCGRP-producing cell types. Nevertheless, the clinical significance of this difference needs to be explored in larger cohort studies in parallel with mechanistic studies that focus on identifying the underlying molecular mechanisms.

aCGRP can be present in a variety of body tissues where it plays tissue-specific activities. The main source of plasma aCGRP is believed to be from the perivascular nerve endings and it is elevated in pathological conditions such as migraine as it is currently being studied as a new therapeutic target [45]. In our study aCGRP levels were elevated at pulmonary exacerbation in patients with positive *S. aureus*. A previous study by El Karim et al., revealed that aCGRP displayed antimicrobial activity against several types of bacteria including *S. aureus* [46].

This was a pilot study, not without limitations. First, is the small sample size that the study could be underpowered in regard to serum VIP and aCGRP levels associations with patients' factors. Second, despite the standard definition of pulmonary exacerbation, treatments were not standardized across different participating sites in this study. Third, Pulmonary function test as represented by FEV1 was only recorded for few patients, so it was not included as a variable in our study and the patient response to therapy was assessed clinically by the responsible physician. The strength of the study however is that the CF patients were analyzed at different stages of the disease.

## Conclusion

We analyzed for the first time the serum levels of VIP and aCGRP in CF patients, and we investigated the longitudinal changes in their levels at pulmonary exacerbation and post-antibiotic therapy. We could only show significant changes in serum aCGRP levels following

treatment of pulmonary exacerbations. However, our present preliminary data provide evidence for the eligibility of serum VIP and aCGRP as systemic markers for pulmonary inflammation and as potential biomarkers for response to exacerbation antibiotic therapy in patients with CF.

## Supporting information

**S1 Data.**
(XLSX)

## Acknowledgments

The authors would like to acknowledge the contribution from the nurses and lab technicians at the hospitals for their help in collecting blood samples from patients.

## Author Contributions

**Conceptualization:** Maha S. Al-Keilani.

**Data curation:** Maha S. Al-Keilani, Samah Awad, Tala Al Shalakhti, Muna M. Dahabreh, Mohammad-Jaafar Ajlony.

**Formal analysis:** Maha S. Al-Keilani, Hanan M. Hammouri, Basima A. Almomani.

**Funding acquisition:** Maha S. Al-Keilani.

**Investigation:** Maha S. Al-Keilani.

**Methodology:** Maha S. Al-Keilani, Hanan M. Hammouri.

**Project administration:** Maha S. Al-Keilani.

**Supervision:** Maha S. Al-Keilani.

**Writing – original draft:** Maha S. Al-Keilani.

**Writing – review & editing:** Maha S. Al-Keilani, Samah Awad, Hanan M. Hammouri, Tala Al Shalakhti, Basima A. Almomani, Muna M. Dahabreh, Mohammad-Jaafar Ajlony.

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
