## [Decision Letter · Decision Letter 0]

1 Nov 2022

PONE-D-22-25130Evaluation of VIP and aCGRP during pulmonary exacerbation in cystic fibrosis patients: a prospective study.PLOS ONE

Dear Dr. Al-Keilani,

Thank you for submitting your manuscript to PLOS ONE. After careful consideration, we feel that it has merit but does not fully meet PLOS ONE’s publication criteria as it currently stands. Therefore, we invite you to submit a revised version of the manuscript that addresses the points raised during the review process.

Authors should pay particular attention to the following points:Application, description and interpretation of all statistical analyses must be revised, reviewers 1 & 2 expressed major concerns on the study design, some methods, statistical data and interpretation.Please address specific changes to introduction and discussion as suggested by reviewer 2 and clarify additional inquires by reviewer 1.Please submit your revised manuscript by Dec 16 2022 11:59PM. If you will need more time than this to complete your revisions, please reply to this message or contact the journal office at plosone@plos.org. Please include the following items when submitting your revised manuscript:A rebuttal letter that responds to each point raised by the academic editor and reviewer(s). You should upload this letter as a separate file labeled 'Response to Reviewers'.A marked-up copy of your manuscript that highlights changes made to the original version. You should upload this as a separate file labeled 'Revised Manuscript with Track Changes'.An unmarked version of your revised paper without tracked changes. You should upload this as a separate file labeled 'Manuscript'.

We look forward to receiving your revised manuscript.

Kind regards,

Santiago Partida-Sanchez

Academic Editor

PLOS ONE

Journal Requirements:

Reviewers' comments:

Reviewer's Responses to Questions

**Comments to the Author**

1. Is the manuscript technically sound, and do the data support the conclusions?

Reviewer #1: No

Reviewer #2: Partly

2. Has the statistical analysis been performed appropriately and rigorously? 

Reviewer #1: No

Reviewer #2: No

3. Have the authors made all data underlying the findings in their manuscript fully available?

Reviewer #1: Yes

Reviewer #2: Yes

4. Is the manuscript presented in an intelligible fashion and written in standard English?

Reviewer #1: Yes

Reviewer #2: Yes

5. Review Comments to the Author

Reviewer #1: The submission by Al-Keilani entitled “Evaluation of VIP and aCGRP during pulmonary exacerbation in cystic fibrosis patients: a prospective study” describes analyses serum concentrations of two potential biomarkers of inflammation from 21 people with cystic fibrosis (CF) who were treated for pulmonary exacerbation.

Major comments.

The authors make a reasonable argument for the potential utility of a systemic biomarker when handling pulmonary exacerbation. However, although their title suggests they have conducted a prospective study, they have not described a prospective hypothesis (that increase or decrease of a given biomarker will be associated with a given outcome) or a corresponding prospective statistical analysis plan (describing study power to detect a response given the anticipated response magnitude, the variance of a given measure, and the size of the population studied). Further, there is no description of how the investigators planned to adjust statistical tests for multiplicity. Thus, although data for this submission were collected prospectively, this is, in fact, a retrospective analysis. As such, it is problematic that statistical test results are reported/interpreted in the abstract for differences in serum biomarker concentrations a) before versus after exacerbation treatment (p>.05), b) by patient sex (p=.038), and c) as a function of Pseudomonas infection (p=.028) without acknowledging type I error inflation resulting from the simultaneous statistical testing of 36 different associations. Given the retrospective nature of these analyses, the investigators should not be reporting P values for tests, but rather providing point estimates and 95% confidence intervals and allowing readers to draw conclusions regarding associations. Given the number of point estimates generated in these analyses, the authors should consider presenting associations as forest plots, which would allow readers to study the juxtapositions confidence intervals for different tests.

Minor comments

The statement that “most of the time patients’ FEV1 values do not improve after antibiotic therapy (8)” which cites D.B. Sander’s 2010 epidemiologic analyses, is an incorrect extrapolation of those data. Sanders and colleagues reported the proportion of individuals with CF who had an FEV1 recorded after exacerbation treatment that was equal to or greater than their maximum FEV1 recorded in the prior year. This was not an analysis of change in FEV1 from immediately prior to immediately after treatment. With respect to the statement about FEV1 improvement with treatment, I refer the authors to the results of a large prospective study of CF exacerbation treatment [Am J Respir Crit Care Med. 2021 Dec 1;204(11):1295-1305], which show that FEV1 does increase after exacerbation treatment for most people with CF.

The use of Fuchs criteria to define exacerbations for inclusion in this analysis should be clarified. As written, it would appear that any individual presenting with any of the 12 Fuchs criteria could be diagnosed with exacerbation and included in the study. In the past, Fuchs criteria have been applied as a means of defining an event as an exacerbation for the purpose of subsequent analyses after a clinician has made the decision to treat a respiratory event with antimicrobials. Further, Fuchs definitions have typically required the presence of at least 4 of 12 criteria.

The authors’ suggestion in the abstract that “Future studies with larger sample size are required to investigate the clinical importance of VIP and aCGRP in cystic fibrosis patients” seems at odds with their results and conclusions. Don’t the presented analyses indicate a lack of clinical correlation for these markers, or do the authors believe that their analyses were substantially underpowered? If the latter is the case, is it appropriate to report these (negative) results?

Reviewer #2: The authors explore changes in levels of VIP and ACGRP in people with CF treated for a pulmonary exacerbations. The sample size is quite small and subjects are predominantly children.

Introduction

- The introduction is long and contains irrelevant material to the topic of the study (e.g. refers to modifier genes), where the topic of the paper is biomarkers. The introduction should be reviewed and made more concise focusing on just background of relevance to the current study.

Methods

- Failure to measure FEV1 would substantially impact on the ability to assess response to treatment of pulmonary exacerbation.

- C- reactive protein is a widely accepted marker of inflammation in CF – Were levels measured in this population. If so, they should be reported and comment made on whether these correlate with VIP / aCGRP

- Mostly children – Mean age 8.7years - BMI probably not best measure of nutrition in children and at the least BMI z-scores should be reported

- No measure of response to antibiotic therapy is reported – e.g. how many patients returned to a symptomatic baseline. Consequently, it is not possible to gauge whether the failure of VIP / aCGRP to change with treatment reflects a lack of association

- No power calculation – It may be possible to determine a necessary sample size to detect a difference in these measured based on the literature in asthma exacerbations

Results

- Means and SEM were reported but a lot of the analysis used non-parametric measures suggesting data was not normally distributed – In which case, Median and IQR (or range) would be preferred.

- Only univariate analysis has been performed and results may be subject to confounding. E.g. it is stated that Female subjects and subjects with P. aeruginosa had higher levels of aCGRP1. But, it is not known if the spread of P. aeruginosa infection was similar between male and female subjects. Multivariate analysis should be performed.

- The small sample size leaves the study open toe Type I and II errors and comment should be made on this.

Discussion

- The discussion is confused. It is unclear whether the authors are proposing that VIP act as a modifier of CFTR function and thereby impacts on disease severity, or VIP can be used as a biomarker of inflammation in CF. As this is a study looking at its function as a biomarker of exacerbations, I would recommend that discussion is limited to it function as a biomarker.

- Line 233 – “Previous reports Indicate VIP and aCGRP may be involved in the pathophysiology of CF exacerbations” – Please provide references.

6. PLOS authors have the option to publish the peer review history of their article (what does this mean?). If published, this will include your full peer review and any attached files.

Reviewer #1: No

Reviewer #2: No

---

## [Author Response · Author response to Decision Letter 0]

17 Feb 2023

Journal Requirements:

Response: Confirmed.

Response: The following sentence was added to the Consent to Participate subsection: “All patients signed a written informed consent at the start of the study. For patients under eighteen years old, the informed consent was signed by one of their parents.”

In the current study, the patients were recruited prospectively, and the data was collected from both patients and their medical records. As stated in the Ethical Approval subsection “The research was performed in accordance with the Declaration of Helsinki, and the protocol was approved by the Research and Ethics Committee at Jordan University of Science and Technology (registration number 34/111/2017, date of approval January 7, 2018), where all methods were performed in accordance with the relevant guidelines and regulations.” 

Response: caption has been added.

Reviewers’ comments

Reviewer #1: 

The submission by Al-Keilani entitled “Evaluation of VIP and aCGRP during pulmonary exacerbation in cystic fibrosis patients: a prospective study” describes analyses serum concentrations of two potential biomarkers of inflammation from 21 people with cystic fibrosis (CF) who were treated for pulmonary exacerbation.

Major comments.

1. The authors make a reasonable argument for the potential utility of a systemic biomarker when handling pulmonary exacerbation. However, although their title suggests they have conducted a prospective study, they have not described a prospective hypothesis (that increase or decrease of a given biomarker will be associated with a given outcome) or a corresponding prospective statistical analysis plan (describing study power to detect a response given the anticipated response magnitude, the variance of a given measure, and the size of the population studied). Further, there is no description of how the investigators planned to adjust statistical tests for multiplicity. Thus, although data for this submission were collected prospectively, this is, in fact, a retrospective analysis. As such, it is problematic that statistical test results are reported/interpreted in the abstract for differences in serum biomarker concentrations a) before versus after exacerbation treatment (p>.05), b) by patient sex (p=.038), and c) as a function of Pseudomonas infection (p=.028) without acknowledging type I error inflation resulting from the simultaneous statistical testing of 36 different associations. Given the retrospective nature of these analyses, the investigators should not be reporting P values for tests, but rather providing point estimates and 95% confidence intervals and allowing readers to draw conclusions regarding associations. Given the number of point estimates generated in these analyses, the authors should consider presenting associations as forest plots, which would allow readers to study the juxtapositions confidence intervals for different tests.

Response: The problem of retesting was solved using repeated measures multiple ANOVA (MANOVA) models. Regarding the univariate effect, we used forest plots for point estimates and 95% confidence intervals.

Minor comments

2. The statement that “most of the time patients’ FEV1 values do not improve after antibiotic therapy (8)” which cites D.B. Sander’s 2010 epidemiologic analyses, is an incorrect extrapolation of those data. Sanders and colleagues reported the proportion of individuals with CF who had an FEV1 recorded after exacerbation treatment that was equal to or greater than their maximum FEV1 recorded in the prior year. This was not an analysis of change in FEV1 from immediately prior to immediately after treatment. With respect to the statement about FEV1 improvement with treatment, I refer the authors to the results of a large prospective study of CF exacerbation treatment [Am J Respir Crit Care Med. 2021 Dec 1;204(11):1295-1305], which show that FEV1 does increase after exacerbation treatment for most people with CF.

Response: The reference has been replaced as recommended.

3. The use of Fuchs criteria to define exacerbations for inclusion in this analysis should be clarified. As written, it would appear that any individual presenting with any of the 12 Fuchs criteria could be diagnosed with exacerbation and included in the study. In the past, Fuchs criteria have been applied as a means of defining an event as an exacerbation for the purpose of subsequent analyses after a clinician has made the decision to treat a respiratory event with antimicrobials. Further, Fuchs definitions have typically required the presence of at least 4 of 12 criteria.

Response: Diagnosis of pulmonary exacerbation was made based on the clinical decision of the pulmonologist. When recruiting patients, we made sure that every patient had at least 4 of the 12 Fuchs criteria to define exacerbation. The Methods section was adjusted for better clarification.

4. The authors’ suggestion in the abstract that “Future studies with larger sample size are required to investigate the clinical importance of VIP and aCGRP in cystic fibrosis patients” seems at odds with their results and conclusions. Don’t the presented analyses indicate a lack of clinical correlation for these markers, or do the authors believe that their analyses were substantially underpowered? If the latter is the case, is it appropriate to report these (negative) results?

Response: In the current study we have limited access to patients, and we have been working on the recruitment for about a year and could only include this number due to lack of follow-up samples. Patients may admit at night, so sometimes it was not feasible to obtain the initial sample, or the patient was discharged early and so could not take the last sample. 

However, with the re-analysis performed. We could find statistically significant results by using the appropriate statistical test. Accordingly, the abstract has been rewritten to represent the new findings. Also we have added the term “pilot” to the title.

Reviewer #2: 

1. The authors explore changes in levels of VIP and aCGRP in people with CF treated for a pulmonary exacerbations. The sample size is quite small and subjects are predominantly children.

Response: In the current study we have limited access to patients, and we have been working on the recruitment for about a year and could only include this number due to lack of follow-up samples. Patients may admit at night, so sometimes it was not feasible to obtain the initial sample, or the patient was discharged early and so could not take the last sample. Thus, this was a consecutive sampling method, and the study sample represents all patients with the three blood samples at the three occasions as described in the methods section, if missing one sample the patient was excluded from the study. 

2. Introduction: the introduction is long and contains irrelevant material to the topic of the study (e.g. refers to modifier genes), where the topic of the paper is biomarkers. The introduction should be reviewed and made more concise focusing on just background of relevance to the current study.

Response: irrelevant sentences focusing on genetics have been deleted as recommended and the introduction has been rephrased accordingly.

3. Methods

A. Failure to measure FEV1 would substantially impact on the ability to assess response to treatment of pulmonary exacerbation.

Response: As we mentioned in the limitations “Pulmonary function test as represented by FEV1 was only recorded for few patients so it was not included as a variable in our study and the patient response to therapy was assessed clinically by the responsible physician.” 

The inability to do pulmonary function test for the majority of the patients and at certain time points only was due to several factors which are described in the introduction section: “However, such testing is not feasible in young children and in patients with severe lung disease and is not accurate in mild conditions where the test is usually normal (7). Besides the fact that several months are required to notice a clinical response after initiation of anti-inflammatory therapy and most of the time patients’ FEV1 values do not improve after antibiotic therapy (8)”.

B. C- reactive protein is a widely accepted marker of inflammation in CF – Were levels measured in this population. If so, they should be reported and comment made on whether these correlate with VIP / aCGRP

Response: C-reactive protein levels were not measured in this study, however, investigating inflammatory markers and correlating them with VIP and aCGRP is of interest and high importance to identify the inflammatory roles of these target markers in CF.

C. Mostly children – Mean age 8.7years - BMI probably not best measure of nutrition in children and at the least BMI z-scores should be reported

Response: As it is mentioned in the method, BMI z-score has been calculated using CDC calculator (Chou JH, Roumiantsev S, Singh R. PediTools Electronic Growth Chart Calculators: Applications in Clinical Care, Research, and Quality Improvement. J Med Internet Res. 2020 Jan 30;22(1):e16204. doi: 10.2196/16204. PMID: 32012066; PMCID: PMC7058170.) as recommended and has been included in the analysis

D. No measure of response to antibiotic therapy is reported – e.g. how many patients returned to a symptomatic baseline. Consequently, it is not possible to gauge whether the failure of VIP / aCGRP to change with treatment reflects a lack of association.

Response: This was a consecutive sampling method, and the study sample represents all patients with the three blood samples at the three occasions as described in the methods section, if missing one sample the patient was excluded from the study. The patient’s response to therapy was assessed clinically by the responsible physician, and the last sample was taken after 2 weeks of therapy, which was the time when the patients returned to a symptomatic baseline.

E. No power calculation – It may be possible to determine a necessary sample size to detect a difference in these measured based on the literature in asthma exacerbations.

Response: It is a pilot study with limited access to patients. We have been working on the recruitment for about a year and could only include this number due to lack of follow-up samples. Patients may admit at night, so sometimes it was not feasible to obtain the initial sample, or the patient was discharged early and so could not take the last sample. The study sample represents all patients with the three blood samples at the three occasions as described in the methods section, if missing one sample the patient was excluded from the study. 

4. Results

A. Means and SEM were reported but a lot of the analysis used non-parametric measures suggesting data was not normally distributed – In which case, Median and IQR (or range) would be preferred.

Response: The biostatistician re-analyzed the data where Q-Q plots were used to verify the normality of VIP and aCGRP variables. There was a need for log transformations to normalize VIP variables, whereas aCGRP variables were assumed to be normally distributed without requiring any transformations. Thus, we are keeping means and SEM as descriptive statistics for continuous variables.

B. Only univariate analysis has been performed and results may be subject to confounding. E.g. it is stated that Female subjects and subjects with P. aeruginosa had higher levels of aCGRP1. But, it is not known if the spread of P. aeruginosa infection was similar between male and female subjects. Multivariate analysis should be performed.

Response: We adjusted the analysis, and we used repeated measures multiple ANOVA (MANOVA) as described in the statistical analysis subsection.

C. The small sample size leaves the study open to Type I and II errors and comment should be made on this.

Response: In the current study we have limited access to patients and thus small sample size. We have been working on the recruitment for about a year and could only include this number due to lack of follow-up samples. Patients may admit at night, so sometimes it was not feasible to obtain the initial sample, or the patient was discharged early and so could not take the last sample. Thus, this was a consecutive sampling method, and the study sample represents all patients with the three blood samples at the three occasions as described in the methods section, if missing one sample the patient was excluded from the study. 

5. Discussion

A. The discussion is confused. It is unclear whether the authors are proposing that VIP act as a modifier of CFTR function and thereby impacts on disease severity, or VIP can be used as a biomarker of inflammation in CF. As this is a study looking at its function as a biomarker of exacerbations, I would recommend that discussion is limited to it function as a biomarker.

Response: The discussion section has been rewritten according to the new results, and more focus on the role of VIP as inflammation biomarker has been further discussed.

B. Line 233 – “Previous reports Indicate VIP and aCGRP may be involved in the pathophysiology of CF exacerbations” – Please provide references.

Response: References have been added as requested.

---

## [Decision Letter · Decision Letter 1]

27 Feb 2023

PONE-D-22-25130R1Evaluation of serum VIP and aCGRP during pulmonary exacerbation in cystic fibrosis: a longitudinal pilot study of patients undergoing antibiotic therapy.PLOS ONE

Dear Dr. Al-Keilani,

Thank you for submitting your manuscript to PLOS ONE. After careful consideration, we feel that it has merit but does not fully meet PLOS ONE’s publication criteria as it currently stands. Therefore, we invite you to submit a revised version of the manuscript that addresses the points raised during the review process.

We look forward to receiving your revised manuscript.

Kind regards,

Santiago Partida-Sanchez

Academic Editor

PLOS ONE

Reviewers' comments:

Reviewer's Responses to Questions

**Comments to the Author**

1. If the authors have adequately addressed your comments raised in a previous round of review and you feel that this manuscript is now acceptable for publication, you may indicate that here to bypass the “Comments to the Author” section, enter your conflict of interest statement in the “Confidential to Editor” section, and submit your "Accept" recommendation.

Reviewer #1: (No Response)

2. Is the manuscript technically sound, and do the data support the conclusions?

Reviewer #1: No

3. Has the statistical analysis been performed appropriately and rigorously? 

Reviewer #1: I Don't Know

4. Have the authors made all data underlying the findings in their manuscript fully available?

Reviewer #1: Yes

5. Is the manuscript presented in an intelligible fashion and written in standard English?

Reviewer #1: Yes

6. Review Comments to the Author

Reviewer #1: The authors of the PLOS ONE submission “Evaluation of serum VIP and aCGRP during pulmonary exacerbation in cystic fibrosis: a longitudinal pilot study of patients undergoing antibiotic therapy” have addressed my technical concerns regarding the use of multiple univariate comparisons. However, there remains an aspect of this submission (which I could not fully appreciate due to the previous analytical approach) that must be resolved.

MAJOR COMMENTS:

The authors engage in a form of circular reasoning regarding the utility of aCGRP and VIP measure as biomarkers of CF exacerbation response:

First, they state that “monitoring of cystic fibrosis is difficult especially in cases of pulmonary exacerbations where patients fail to regain the baseline pulmonary function despite standard treatment” (its not clear how a treating clinician can know that an individual has failed to regain baseline without monitoring… so clearly monitoring must be possible in this instance).

Then, they proceed to describe aCGRP and VIP changes during antimicrobial exacerbation treatment as a biomarker of treatment response without methodologic description of how *clinical* responses to treatment were assessed, recorded, or incorporated into analyses. They state that spirometry was either not available or uninformative and thus they relied on clinician evaluation… but I could not find these clinician assessments of response. How can the investigators conclude that aCGRP changes are indicative of treatment response without describing responses?

MINOR COMMENTS

The authors can be more disciplined in their manuscript construction. Lines 167 to 170 on page 9 of the results contain elements of Methods, Results and Discussion: “By analyzing the changes overtime of LOG VIP and aCGRP2 serum levels using repeated measures models, we investigated the factors responsible for these changes [a Method]. It was found that there were significant between-subjects effects over the LOG VIP [a Result], which indicates that the difference between the levels means which is not due to the time [a Discussion].”

Further, rather than state: “Figure 3 shows the LS means 171 and p-values for these effects” after this passage, why not just append the second sentence with “(Figure 3)”?

This manuscript is overly long at the expense of a clear description of observations. There is entirely too much discussion of the biology of aCGRP and VIP. A few sentences identifying and referencing key observations is more than adequate. Observations that these are interesting molecules are tangential to the matter at hand, which is how well they associate with exacerbation treatment.

The figures are almost illegible. The fonts are too small, especially in Figures 3 and 4 and the point estimates for LS means in Figures 1 and 2 should have 95% CI or SE bars included.

7. PLOS authors have the option to publish the peer review history of their article (what does this mean?). If published, this will include your full peer review and any attached files.

Reviewer #1: No

---

## [Author Response · Author response to Decision Letter 1]

14 Mar 2023

Reviewer #1:

The authors of the PLOS ONE submission “Evaluation of serum VIP and aCGRP during pulmonary exacerbation in cystic fibrosis: a longitudinal pilot study of patients undergoing antibiotic therapy” have addressed my technical concerns regarding the use of multiple univariate comparisons. However, there remains an aspect of this submission (which I could not fully appreciate due to the previous analytical approach) that must be resolved.

Response: A multivariate model with stepwise selection was used instead of the univariate model to test all variables and select those with significance without retesting or inflating alpha levels.

MAJOR COMMENTS:

The authors engage in a form of circular reasoning regarding the utility of aCGRP and VIP measure as biomarkers of CF exacerbation response:

First, they state that “monitoring of cystic fibrosis is difficult especially in cases of pulmonary exacerbations where patients fail to regain the baseline pulmonary function despite standard treatment” (it’s not clear how a treating clinician can know that an individual has failed to regain baseline without monitoring… so clearly monitoring must be possible in this instance).

Then, they proceed to describe aCGRP and VIP changes during antimicrobial exacerbation treatment as a biomarker of treatment response without methodologic description of how *clinical* responses to treatment were assessed, recorded, or incorporated into analyses. They state that spirometry was either not available or uninformative and thus they relied on clinician evaluation… but I could not find these clinician assessments of response. How can the investigators conclude that aCGRP changes are indicative of treatment response without describing responses?

Response: Thanks for your comment. We agree that the statement “Adequate monitoring of cystic fibrosis is difficult especially in cases of pulmonary exacerbations where patients fail to regain the baseline pulmonary function despite standard treatment” was not correct and we deleted it. We replaced the statement by the following: “Objective monitoring of improvement during treatment of pulmonary exacerbation can be difficulty in children when pulmonary function testing cannot be obtained.”

It was stated in the Method section lines 110-112 that the duration of treatment was 14 days, after which patients returned to baseline: 

“The duration of antibiotic therapy for pulmonary exacerbation was 2 weeks for all patients, which is the duration of antibiotic therapy where patients returned to a symptomatic baseline”, and then we added the phrase “and had clinically improved based on clinician’s assessment” to make it clearer that the improvement was based on clinical response.

MINOR COMMENTS

The authors can be more disciplined in their manuscript construction. 

1. Lines 167 to 170 on page 9 of the results contain elements of Methods, Results and Discussion: “By analyzing the changes overtime of LOG VIP and aCGRP2 serum levels using repeated measures models, we investigated the factors responsible for these changes [a Method]. It was found that there were significant between-subjects effects over the LOG VIP [a Result], which indicates that the difference between the levels means which is not due to the time [a Discussion].”

Response: The sentence “By analyzing the changes overtime of LOG VIP and aCGRP2 serum levels using repeated measures models, we investigated the factors responsible for these changes” has been shifted to the [methods] section, and the sentence “which indicates that the difference between the levels means which is not due to the time” has been shifted to the [discussion] section.

2. Further, rather than state: “Figure 3 shows the LS means 171 and p-values for these effects” after this passage, why not just append the second sentence with “(Figure 3)”?

Response: The paragraph has been rephrased.

3. This manuscript is overly long at the expense of a clear description of observations. There is entirely too much discussion of the biology of aCGRP and VIP. A few sentences identifying and referencing key observations is more than adequate. Observations that these are interesting molecules are tangential to the matter at hand, which is how well they associate with exacerbation treatment.

Response: Changes to the discussion has been performed where extra information about the molecular role of the markers has been deleted, and additional evidence of their role as potential biomarkers in CF has been discussed; lines 240-243 and lines 266-274.

4. The figures are almost illegible. The fonts are too small, especially in Figures 3 and 4 and the point estimates for LS means in Figures 1 and 2 should have 95% CI or SE bars included.

Response: The figures for all factors have been updated. for overall LS means, we have included the SE in tables instead of adding them to the figures.

---

## [Decision Letter · Decision Letter 2]

3 Apr 2023

Evaluation of serum VIP and aCGRP during pulmonary exacerbation in cystic fibrosis: a longitudinal pilot study of patients undergoing antibiotic therapy.

PONE-D-22-25130R2

Dear Dr. Al-Keilani,

We’re pleased to inform you that your manuscript has been judged scientifically suitable for publication and will be formally accepted for publication once it meets all outstanding technical requirements.

Kind regards,

Santiago Partida-Sanchez

Academic Editor

PLOS ONE

Additional Editor Comments (optional):

Reviewers' comments:

Reviewer's Responses to Questions

**Comments to the Author**

1. If the authors have adequately addressed your comments raised in a previous round of review and you feel that this manuscript is now acceptable for publication, you may indicate that here to bypass the “Comments to the Author” section, enter your conflict of interest statement in the “Confidential to Editor” section, and submit your "Accept" recommendation.

Reviewer #1: All comments have been addressed

2. Is the manuscript technically sound, and do the data support the conclusions?

Reviewer #1: Yes

3. Has the statistical analysis been performed appropriately and rigorously? 

Reviewer #1: Yes

4. Have the authors made all data underlying the findings in their manuscript fully available?

Reviewer #1: Yes

5. Is the manuscript presented in an intelligible fashion and written in standard English?

Reviewer #1: Yes

6. Review Comments to the Author

Reviewer #1: The authors have addressed my concerns, although I remain skeptical that a systemic biomarker can supplant physical exam and communication with patients regarding treatment response.

7. PLOS authors have the option to publish the peer review history of their article (what does this mean?). If published, this will include your full peer review and any attached files.

Reviewer #1: No

---

## [Editor Report · Acceptance letter]

27 Apr 2023

PONE-D-22-25130R2 

Evaluation of serum VIP and aCGRP during pulmonary exacerbation in cystic fibrosis: a longitudinal pilot study of patients undergoing antibiotic therapy. 

Dear Dr. Al-Keilani:

I'm pleased to inform you that your manuscript has been deemed suitable for publication in PLOS ONE. Congratulations! Your manuscript is now with our production department. 

Kind regards, 

on behalf of

Professor Santiago Partida-Sanchez 

Academic Editor

PLOS ONE